# An Overview of the Pharmacokinetics and Pharmacodynamics of Landiolol (an Ultra-Short Acting β1 Selective Antagonist) in Atrial Fibrillation

**DOI:** 10.3390/pharmaceutics16040517

**Published:** 2024-04-08

**Authors:** Mariana Floria, Alexandru Florinel Oancea, Paula Cristina Morariu, Alexandru Burlacu, Diana Elena Iov, Cristina Petronela Chiriac, Genoveva Livia Baroi, Celina Silvia Stafie, Magdalena Cuciureanu, Viorel Scripcariu, Daniela Maria Tanase

**Affiliations:** 1Department of Internal Medicine, Faculty of Medicine, “Grigore T. Popa” University of Medicine and Pharmacy, 700115 Iasi, Romania; floria.mariana@umfiasi.ro (M.F.); alexandru.burlacu@umfiasi.ro (A.B.); diana-elena.iov@d.umfiasi.ro (D.E.I.); daniela.tanase@umfiasi.ro (D.M.T.); 2Saint Spiridon Emergency Hospital, 700115 Iasi, Romania; chiriac.cristina@spitalspiridon.ro (C.P.C.); livia.baroi@umfiasi.ro (G.L.B.); 3Cardiovascular Disease Institute, 700503 Iasi, Romania; 4Department of Surgery, Faculty of Medicine, “Grigore T. Popa” University of Medicine and Pharmacy, 700115 Iasi, Romania; viorel.scripcariu@umfiasi.ro; 5Department of Preventive Medicine and Interdisciplinarity, Faculty of Medicine, “Grigore T. Popa” University of Medicine and Pharmacy, 700115 Iasi, Romania; celina.stafie@umfiasi.ro; 6Department of Pharmacology, “Grigore T. Popa” University of Medicine and Pharmacy, 700115 Iasi, Romania; mag.cuciureanu@umfiasi.ro; 7Regional Institute of Oncology, 700483 Iasi, Romania

**Keywords:** landiolol, atrial fibrillation, beta-blockers, perioperative tachyarrhythmias, cardiac surgery, arrhythmias

## Abstract

Landiolol is an ultra-short-acting, selective β1-adrenergic receptor blocker that was originally approved in Japan for the treatment of intraoperative tachyarrhythmias. It has gained attention for its use in the management of tachyarrhythmias and perioperative tachycardia, especially atrial fibrillation for both cardiac and non-cardiac surgeries. It can be the ideal agent for heart rate control due to its high β1-selectivity, potent negative chronotropic effect, a limited negative inotropic potential, and an ultrashort elimination half-life (around 4 min); moreover, it may have a potential therapeutic effects for sepsis and pediatric patients. Landiolol seems to be superior to other short-acting and selective beta-blockers such as esmolol. This review aims to provide a comprehensive overview of landiolol, a new ultra-short-acting β1 selective antagonist, including its pharmacology, clinical applications, efficacy, safety profile, and future directions in research and clinical data.

## 1. Introduction

Tachyarrhythmias, the most frequent being atrial fibrillation (AF), may bring significant risks to patients, especially those with underlying cardiovascular conditions. While various β-blockers options exist, the need for a rapid and effective intervention has led to the development of ultra-short-acting β-blocker such as landiolol. This review will explore the pharmacological properties, clinical applications, potential benefits, challenges, and perspectives of landiolol in the management of tachyarrhythmias. 

Atrial fibrillation, whose mechanism consists of a concept using Coumel’s Triangle that requires a trigger for initiation, a catalyst agent, and an arrhythmogenic substrate for the perpetuation and maintenance of the trigger, increases heart rate (HR) and oxygen consumption, and implicates the loss of atrial systole, which will bring a poorer prognosis for frail patients, such as perioperative patients. This required the study and development of a β-blocker such as landiolol, which has been primarily studied and utilized in the management of tachyarrhythmias, not only AF but also atrial flutter (AFL), because of its rapid onset and offset of action. In this way, it may be the best option for acute situations, where immediate HR control is required [1,2,3,4,5,6,7]. 

Several clinical studies have demonstrated the efficacy of landiolol in controlling HR in patients with tachyarrhythmias. Its rapid onset of action and short half-life allow for precise titration to control HR control without the risk of prolonged β-blockade. Landiolol has shown favorable safety and tolerability profiles, with minimal negative inotropic effects, making it suitable for patients with compromised cardiac function. Compared with esmolol, another intravenous β-blocker, the high β1 selectivity (β1/β2 ratio of 255:1) of landiolol allows it to produce a more rapid HR decrease, avoiding in the same time the reductions in mean arterial blood pressure (BP). This is the reason landiolol has also been found useful in left ventricular dysfunction patients and fatal arrhythmia requiring emergency treatment [8,9,10,11,12,13].

A meta-analysis showed that landiolol is effective in the prevention of AF after cardiac surgery (especially coronary artery bypass grafting) and without increasing the risk of major complications. Also, this β-blocker can manage postoperative atrial fibrillation (POAF) in non-cardiac surgeries, such as esophagectomy. Ojima et al. showed that administration of landiolol 3 μg/kg/min from the first postoperative day in patients who had an esophagectomy may reduce the incidence of POAF (landiolol: 30% vs. placebo: 10%, *p* = 0.012) and postoperative complications (landiolol: 40% vs. placebo: 60%, *p* = 0.046), and even lead to a reduction in IL-6 cytokine, which is a promotor for inflammation [5,6,14,15,16,17].

Regarding percutaneous coronary intervention, another study showed that using 3 μg/kg/min may be an independent predictor of an ST-segment resolution, and may prevent Killip class grade progression without important cardiac complications, such as severe bradycardia, hypotension, or atrioventricular block [15,18,19].

In septic patients, it is known that tachycardia is associated with a poor outcome, and the use of a catecholamine vasopressor may increase cardiac adverse events; this is why a β-adrenergic blocker could significantly decrease mortality, by preserving HR and heart load. Landiolol can be the ideal solution in this case, especially for its ability to decrease serum levels of TNF-α and IL-6 in rat models, according to Seki et al. [7,9,11,16,20,21].

Landiolol may also be used to control HR in pediatric patients with non-arrhythmic sinus tachycardia, heart failure, and pulmonary hypertension. According to Schroeder et al., landiolol administration is well tolerated and safe in pediatric patients, being able to achieve a quick HR control; it is associated with an improvement in ventricular dysfunction and pulmonary hypertension severity in critically ill infants [22].

## 2. Overview of Pharmacodynamics and Pharmacokinetics in Comparison with Other β-Blockers

Landiolol hydrochloride is an ultra-short-acting β1-adrenergic receptor blocker with a rapid onset and offset of action. It exerts its effects by competitively blocking β1-adrenergic receptors, leading to a reduction in HR, myocardial contractility, and conduction velocity. It specifically targets the beta-1 adrenergic receptors, which are predominantly found in the myocardium, in this way competing with endogenous catecholamines (epinephrine and norepinephrine). Following the blockade of β-adrenergic receptors, several intracellular molecules and signaling pathways are affected (as shown in Figure 1):G protein-coupled receptor signaling: β-1 adrenergic receptors are G protein-coupled receptors, meaning that their activation leads to the activation of intracellular signaling pathways through G proteins. When catecholamines (e.g., epinephrine and norepinephrine) bind to β-1 receptors, they activate stimulatory G proteins, leading to the production of cyclic adenosine monophosphate (cAMP) from adenosine triphosphate by the enzyme adenylyl cyclase;cAMP production: Cyclic AMP, a second messenger, plays a central role in mediating the effects of β-adrenergic receptor activation. Increased levels of cAMP lead to the activation of protein kinase A (PKA), a key enzyme involved in regulating cellular function. PKA phosphorylates a variety of target proteins, leading to changes in cellular function and gene expression, one of the major intracellular effects of β-1 receptor activation being the regulation of intracellular calcium levels in cardiac myocytes. The activation of β-1 receptors leads to an increase in intracellular calcium levels through several mechanisms, including enhanced calcium release from the sarcoplasmic reticulum and increased calcium entry through L-type calcium channels in the cell membrane. The increase in intracellular calcium levels and the subsequent activation of PKA through the ryanodine receptor 2, sarcoplasmic/endoplasmic reticulum calcium ATPase 2 and phospholamban lead to enhanced contractility and increased HR in cardiac myocytes [3,13,17,23].

**Figure 1 pharmaceutics-16-00517-f001:**
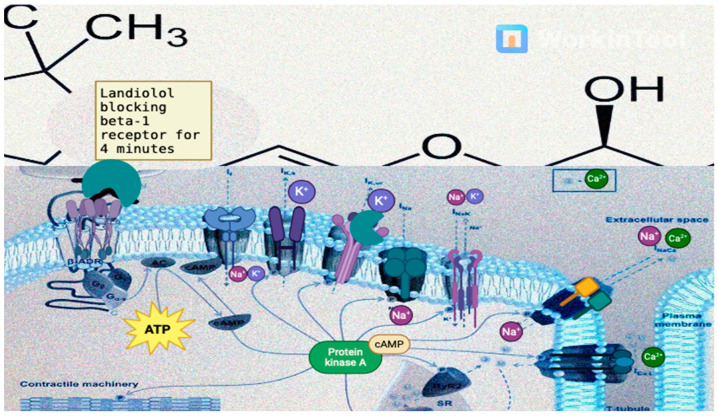
The mechanism of action for Landiolol. This image was created with Biorender.com (accessed on 23 March 2024). β–ADR: β adrenergic receptor; I_Na_: sodium current; I_Ks_: slow delayed rectifier potassium current; If: funny current; I_NaK_: sodium–potassium pump current; cAMP: cyclic adenosine monophosphate; RyR2: ryanodine receptor 2; ATP: adenosine triphosphate; AC: adenylyl cyclase.

When β-blockers like landiolol block β-1 receptors, they interfere with all of these intracellular signaling pathways. Specifically, the inhibition of β-1 adrenergic receptors leads to reduced cAMP production, the decreased activation of PKA, and altered calcium handling in cardiac myocytes, which will create (detailed in Table 1) [24]: A negative bathmotropic effect—decreased cellular excitability (by inhibiting the effects of catecholamines on beta adrenergic receptors, which reduces the concentration of Ca^2+^ ions in the cytoplasm);A negative dromotropic effect (it decreases the speed of impulse conduction through the atrioventricular node by blocking IKs channels most notably);A negative chronotropic effect (blocking the β-1 receptors leading to a decrease in Ca^2+^ influx into cardiomyocytes during action potential, which is thus responsible for the negative inotropic effect);A negative inotropic effect—Due to its blockade of the β-1 receptors in the myocardium and in this way blocking the release of calcium, necessary for cardiomyocyte contraction;An antiarrhythmic effect—Due to its suppression of triggered activity and the prolongation of a refractory period by blocking IKs channels. In this way, it may help prevent the re-entry of electrical impulses that can lead to certain types of arrhythmias such as AF, AFL, atrioventricular re-entry tachycardia, or atrioventricular nodal reentry tachycardia [25].

**Table 1 pharmaceutics-16-00517-t001:** The impacts of beta-blockers upon If, IKs, INa, and INaK channels.

Current Channel	If	IKs	INa	INaK
Role	A mixed cation current is carried by both Na^+^ and K^+^. It is mainly involved in the pacemaker activity of the sinoatrial node, contributing to the diastolic depolarization and spontaneous firing of action potentials.	A delayed rectifier potassium current plays a role in repolarizing the cardiac action potential. It contributes to the plateau phase of the action potential in cardiac myocytes.	The fast inward sodium current is responsible for the rapid depolarization phase of the cardiac action potential. It plays a critical role in initiating and propagating action potentials in cardiac myocytes.	The sodium–potassium pump (Na^+^/K^+^-ATPase) plays a crucial role in maintaining the resting membrane potential of cardiac myocytes by actively transporting 3 atoms of sodium out of the cell and 2 atoms of potassium into the cell.
The influence of landiolol ^1^	Indirect—by blocking β-1 receptors, landiolol reduces the stimulatory effects of endogenous catecholamines, leading to a decrease in If activity and a subsequent decrease in heart rate.	May have minor effects on IKs, primarily through downstream signaling pathways influenced by the blockade of β-1 adrenergic receptors.	Indirect—by blocking β-1 receptors, landiolol reduces the stimulatory effects of catecholamines on INa, leading to a decrease in the rate of rise of the action potential and a reduction in myocardial excitability.	Not well-documented, its effects on intracellular calcium levels and ion handling in cardiac myocytes, mediated through β-1 receptor blockade, may indirectly influence the activity of the sodium–potassium pump.

^1^ available for all beta-blockers. If: funny current; IKs: slow delayed rectifier potassium current; INa: sodium current; INaK: sodium–potassium pump current.

Compared to other β-blockers, landiolol has a unique pharmacokinetic profile, characterized by its short half-life of approximately 4 min, which allows for the rapid titration and termination of its effects. It distinguishes it from other β-blockers in terms of pharmacodynamics and pharmacokinetics (detailed in Table 2).

Regarding pharmacodynamics, it is more cardio-selective than many other β-blockers, having a greater affinity for β-1 adrenergic receptors relative to β-2 receptors. This selectivity may result in a decreased risk of bronchoconstriction in patients with asthma or chronic obstructive pulmonary disease compared to non-selective β-blockers. Then, it has a rapid onset and short duration of action making it suitable for rapid titration and discontinuation, as needed. This is in contrast to other β-blockers, which often have longer half-lives and durations of action. 

Regarding *pharmacokinetics*, landiolol undergoes rapid metabolism by esterase in the tissues rather than by the liver, leading to its short half-life (about 4 min). This rapid metabolism allows for quick clearance from the body, making it suitable for use in situations where rapid titration or discontinuation is necessary. It is primarily eliminated renally, and its rapid clearance contributes to its short duration of action. Other β-blockers, such as metoprolol and propranolol, are predominantly eliminated through hepatic metabolism and have longer half-lives [26,32,33,34]. 

Recent studies have also shown the anti-ischemic, anti-inflammatory, antioxidant, and cardioprotective effects of landiolol, as they claim significantly decreased levels of biomarkers for both inflammation (high-sensitive C-reactive protein, pentraxin-3 and asymmetric dimethylarginine) and for myocardial ischemia (creatine kinase isoenzyme MB, troponin-I and human heart fatty acid binding protein) in patients treated with this betablocker. While various factors have been reported to be associated with AF after cardiac surgery, such as cardiac dysfunction, inflammation, ischemia–reperfusion injury, oxidative stress, fibrosis, and sympathetic hypertonia, the PASCAL trial sustains that landiolol has an anti-ischemic action, an anti-inflammatory action and sympatholytic effects, as it showed that CPK-MB, troponin-I, and heart fatty acid-binding protein were significantly lower in the landiolol group compared to the control group, suggesting that the prevention of ischemia might have reduced the occurrence of AF after CABG. The PLATON trial showed that AF occurred in 3 patients (10%) in the landiolol group versus 12 (40%) in the control group after cardiac surgery, and that during the early postoperative period, levels of brain natriuretic peptide and ischemic biomarkers were significantly lower in the landiolol group than the control group. The BABYLON trial claims that in the group receiving oral bisoprolol postoperatively together with landiolol, high-sensitive C-reactive protein, pentraxin-3 and asymmetric dimethylarginine were all significantly lower in the postoperative phase than in the other two groups (landiolol group and control group without beta-blocker therapy) [32,33,34].

Another study claims the ability to attenuate acute lung injury in a rat model of endotoxin-induced sepsis. Cotreatment with landiolol provided protection against lung injury and reduced the wet-to-dry weight ratio in the lung in a rat model of lipopolysaccharide-induced sepsis. Moreover, the treatment with landiolol was associated with a significant reduction in serum levels of high-mobility group box 1 (HMGB1), citokynes such as IL-6, and TNF-alpha, as well as leading to less histological lung damage [35].

Landiolol stands out from other β-blockers due to its rapid onset, short duration of action, cardioselectivity, rapid metabolism, and renal elimination. These unique pharmacodynamics and pharmacokinetic properties, including other properties such as anti-ischemic, anti-inflammatory, and antioxidant effects, make it a valuable option in certain clinical scenarios, such as for the management of acute tachyarrhythmias or perioperative tachycardia such as POAF. 

## 3. The Use of Landiolol in Clinical Practice 

Landiolol, with a half-life of 3–4 min, is a β1 blocker characterized as an ultra-short-acting agent. It has received approval in Japan for addressing sinus tachycardia, AFL, and AF. In terms of cardioselectivity, landiolol hydrochloride demonstrates a high degree (β1/β2 = 255) in comparison to esmolol (β1/β2 = 33) or propranolol (β1/β2 = 0.68). It exhibits approximately nine times greater potency in beta-blocking activity in vivo and eight times higher cardioselectivity in vitro compared to esmolol. Landiolol could be an appealing substitute for long-acting β-blockers in both the perioperative environment and among critically ill patients [26].

### 3.1. Postoperative Atrial Fibrillation Prevention and Treatment in Cardiac Surgery

Atrial fibrillation is among the most common complication following heart surgery, with an estimated incidence ranging from 16% to 85% [34]. This potentially life-threatening complication is linked to heightened risks of adverse neurological events, congestive heart failure, perioperative heart failure, myocardial infarction, prolonged length of hospitalization, and higher hospital costs [36]. It poses challenges in the postoperative period and has an impact on the long-term mortality of patients [34]. Sympathetic activation or an exaggerated response to adrenergic stimulation serves as a significant trigger for POAF [36]. β-blockers play a crucial role in both preventing and treating POAF [36]. In this regard, the European Society of Cardiology, in conjunction with the American Heart Association, the American College of Cardiology, and the American College of Chest Physicians, advocates employing β-blockers to prevent AF in patients undergoing cardiac surgery [37,38]. 

One meta-analysis that incorporated thirty-three studies and encompassed a total of 4698 participants who underwent cardiac surgery examined the impacts of β-blockers on POAF. These results indicate a noteworthy reduction in POAF within the group that received β-blockers (16.3%) compared to the placebo group (31.7%) (OR 0.33; 95% CI 0.26 to 0.43; I2 = 55%) [39]. Moreover, in another recent systematic review involving 688 patients, it was revealed that administering β-blockers before elective cardiac surgery resulted in a notable reduction in the risk of POAF. Additionally, this preoperative intervention had an impact on the risk of stroke and all-cause mortality [40]. 

Recent research indicates that the intravenous administration of an ultra-short-acting β-blocker, like landiolol, is associated with a reduced likelihood of developing AF following cardiac surgery. Existing studies have been structured to commence landiolol administration at various points: at the start of anesthesia induction, during surgery, and post-procedure. Nevertheless, there is a lack of research emphasizing the significance of the timing of administration [41]. 

The PASCAL trial was undertaken to assess the effectiveness of intravenous landiolol treatment in preventing POAF. It was a randomized, double-blind, placebo-controlled study. Patients scheduled for coronary artery bypass grafting (CABG) were randomized into two groups: one receiving landiolol during surgery and the other receiving a placebo. Landiolol infusion was started at 2 μg/kg/min during central anastomosis and discontinued after 48 h, with heart rate monitoring and pacing if necessary. Within the landiolol group, AF occurred in 7 patients (5%), whereas in the placebo group, the incidence of AF was observed in 24 patients (34.3%) [33]. Furthermore, the group treated with landiolol exhibited a decrease in inflammatory and ischemic parameters. It can be concluded that the inhibition of sympathetic hypertonia by landiolol led to a reduction in the incidence rate of AF [33]. 

The PELTA study, conducted from 2010 to 2014, involved 150 consecutive patients aged over 70 who underwent cardiovascular surgery for valvular, ischemic heart disease, and aortic disease. These patients were divided into three groups: Group 1 (landiolol at 1 μg/kg/min), Group 2 (landiolol at 2 μg/kg/min), and the control group (no landiolol). Landiolol hydrochloride was administered intravenously once the patient arrived in the intensive care unit (ICU) for a period of 4 days postoperatively. Oral administration of beta-blockers was banned during the study. Continuous electrocardiography monitoring took place throughout the study, and cardiologists subsequently assessed whether AF occurred postoperatively. AF was observed in 24.4% of patients in the control group, 18.2% of patients in Group 1, and 11.1% of patients in Group 2. The conducted statistical analysis revealed a clear correlation between the incidence of POAF and the dosage of landiolol. This suggests that a higher dose of landiolol might be more efficacious in preventing the occurrence of AF following cardiac surgery [42]. 

Studies assessing the effectiveness of landiolol in preventing AF after cardiovascular surgery have been conducted across various patient profiles. One such study, the PLATON study, involved 60 patients with a preoperative left ventricular ejection fraction of less than 35% who had cardiac surgery performed with cardiopulmonary bypass [32]. Patients were randomized into two groups before surgery. In the group receiving landiolol, the administration of landiolol hydrochloride commenced at a dosage of 2 μg/kg/min upon weaning from cardiopulmonary bypass and persisted for a minimum of 2 days. Subsequently, an oral beta-blocker was initiated following the commencement of oral intake. If the oral beta-blocker was initiated on the third postoperative day, the infusion rate of landiolol was reduced to 1 μg/kg/min, and then discontinued after an hour. Conversely, if the oral beta-blocker was not initiated by the third postoperative day, the infusion rate of landiolol was reduced to 1 μg/kg/min once the oral beta-blocker was actually administered, and then discontinued after an hour. The maximum duration for landiolol infusion was limited to 5 days. Among these patients, AF occurred in 10% of those treated with landiolol, as opposed to 40% in those without landiolol (*p* = 0.002). The study proposes that while it is currently recommended to administer an oral beta-blocker within 24 h after surgery, this approach might not be as effective in preventing POAF compared to using landiolol infusion. Additionally, the study group exhibited lower levels of brain natriuretic peptide and ischemic biomarkers, along with a shorter duration of hospitalization. The study concludes that utilizing a low dose (1 μg/kg/min) of landiolol is both safe and beneficial for patients with left ventricular dysfunction. This approach proves effective in preventing episodes of AF that may occur after cardiac surgery [32]. 

A meta-analysis conducted by Hao et al. affirms the positive impact of landiolol in preventing AF in patients undergoing cardiac surgery. Furthermore, the analysis notes favorable effects in terms of fewer adverse events associated with landiolol use. Additionally, the administration of landiolol to prevent AF after cardiac surgery was found to reduce mortality in comparison to using landiolol for treating AF after cardiac surgery [43]. 

When compared to other active substances, landiolol was found to be more effective and safer for administration in patients with POAF. A study conducted in Japan investigated the effects of landiolol compared to diltiazem in patients who developed AF after heart surgery. Out of a total of 335 patients, 71 experienced AF. The conversion to sinus rhythm (SR) occurred in 54.3% of patients treated with intravenous landiolol, in contrast to 30.6% of patients treated with diltiazem. Furthermore, hypotension and bradycardia, as side effects, were less common in the group of patients receiving landiolol [44].

One recent meta-analysis identified the combination of landiolol and bisoprolol as the most effective in reducing the incidence of POAF. Optimal management appears to involve initiating oral beta-blockers before surgery and commencing landiolol either during surgery or in the ICU [41,45]. Previous research by Sezai et al. found that 57% of POAF cases occurred after the completion of 48 h of landiolol infusion [34,41]. Similarly, Yoshioka et al. observed that combining landiolol with carvedilol was more effective than using landiolol alone (limited to 2 days post-surgery), with fewer cases of POAF occurring on days 3–4 in the landiolol/carvedilol group compared to the landiolol-only group [41,46].

In all the above studies, landiolol was well tolerated, with minimal adverse effects, and a low incidence of hypotension, bradycardia, or respiratory symptoms, which did not require stopping treatment or lowering the dose. Thus, landiolol proved to be a β-blocker with a well-established safety profile [47]. 

### 3.2. Postoperative Atrial Fibrillation Prevention and Treatment in Non-Cardiac Surgery

The reasons behind AF occurrence after thoracic or gastrointestinal surgery remain uncertain, but potential contributors may involve the combined impact of heightened vagal tone, hypoxemia, pulmonary hypertension, atrial inflammation, and right heart dilation. Risk factors for POAF encompass factors related to both the patient and the surgical procedure itself, with AF carrying the potential to give rise to additional severe complications [14,48]. 

Individuals undergoing lung resection and experiencing AF also face additional challenges related to pulmonary complications, such as acute respiratory failure or pneumonia. These complications contribute to an elevated risk of both mortality and morbidity [49]. Additionally, POAF extends the duration of hospitalization, accompanied by an associated rise in costs [48]. There is a limited number of studies assessing the efficacy of landiolol in preventing [14,50,51,52,53] and treating [48,49,54,55] AF following non-cardiac surgery. 

Due to its very short action time and significant cardioselectivity, with minimal action on alpha receptors, landiolol can be used in patients with respiratory diseases [56]. Considering these factors, Nakano et al. explored the utility and effectiveness of using landiolol in cases of AF following pulmonary resection. The infusion of landiolol commenced following the identification of supraventricular tachycardia on an electrocardiogram. The continuous intravenous infusion of landiolol resulted in a reduction in HR in these individuals from 135 ± 24 bpm to 85 ± 19 bpm (*p* < 0.0001). Moreover, landiolol administration resulted in conversion to sinus rhythm in 56% of patients [48] (Table 3). 

Two other studies exploring the impact of landiolol hydrochloride on treating AF after thoracic surgery have yielded consistent findings. Patients who underwent landiolol treatment demonstrated significantly diminished decreases in HR, with some experiencing restoration of SR [41,42].

In another study, the effectiveness of landiolol treatment was assessed in comparison to the administration of digoxin and verapamil for addressing AF following pulmonary resection. The study revealed a significant reduction in HR within the landiolol-treated group compared to the control group. Additionally, the time taken for the restoration of SR was significantly shorter in the landiolol-treated group. Furthermore, there were no statistically significant differences in BP between the two groups, and no need for the discontinuation of treatment due to adverse effects was observed [49,54,55].

Landiolol has also been used successfully to prevent the onset of AF after non-cardiac surgery. In a double-blind, randomized trial involving 100 cancer patients undergoing esophageal surgery, landiolol was administered after surgery for the prevention of AF. In patients receiving landiolol for prophylaxis, AF occurred in 5 (10%), whereas in the control group, 15 patients (30%) experienced AF. Moreover, in the landiolol group, HR was significantly lower than in the placebo group [57]. Comparable outcomes were noted in patients undergoing lung resection. Administering a low dose of landiolol at 5 μg/kg/min before the induction of general anesthesia and maintaining it until discharge from the intensive care unit led to a notable reduction in HR. Notably, in this study, the initiation of AF was entirely prevented, with no patients in this group experiencing this tachyarrhythmia [50]. 

Okita et al. conducted a study comparing a group that received landiolol before lung resection surgery to a control group without AF prophylaxis. The incidence of AF after landiolol prophylaxis was 5.2%, while the control group had a higher percentage at 14.2%. Risk factors linked to AF occurrence included older age, ischemic heart failure, longer surgery duration, and lymph node removal. Importantly, the administration of landiolol did not lead to a significant reduction in BP or marked bradycardia [53]. 

### 3.3. Treatment of Atrial Fibrillation in Patients with Cardiac Dysfunction 

Heart failure (HF) and AF frequently coexist, contributing to the deterioration of cardiac function and jeopardizing hemodynamic stability. Acute HF results from the loss of atrial contraction, a rapid ventricular contraction rate, left ventricular underfilling, and increased left atrial pressures [58]. The prevalence of AF in patients with acute HF ranges from 25% to 40%. Its presence exacerbates symptoms, resulting in adverse outcomes, prolonged hospital stays, and increased mortality [58]. 

To address the necessity for HR control, the guidelines for HF recommend β-blocker treatment for patients with reduced ejection fraction (EF) or mild reduced EF [59]. Landiolol has been shown to be an interesting option in patients with cardiac dysfunction and AF [58]. The Japanese Circulation Society advocates the use of landiolol, along with bisoprolol or carvedilol, as the primary treatment for controlling HR in AF (Class IB), for patients with HF, particularly in cases without an accessory pathway [60,61]. 

In a prospective, multicenter study comparing landiolol to digoxin (J-Land study), it was observed that a dosage ranging from 1 to 10 mcg/kg/min of landiolol proved more effective in patients with AF or AFL and an EF of 25–50% compared to digoxin. In nearly half of the patients, a decrease in HR exceeding 20% and achieving an HR below 110 bpm were accomplished within 2 h following the administration of landiolol. In contrast, the digoxin group exhibited this therapeutic success in only 13.9% of patients. This reference study found no correlation between dosage and effects, highlighting that the optimal dosage varied among participants, each exhibiting a variable response to treatment. Consequently, the study underscores the necessity of adjusting the dosage based on the patient’s overall health status and cardiac function [62]. 

A recent study conducted retrospectively across multiple medical centers examined 39 patients with HF with left ventricle ejection fraction (LVEF) between 34 ± 16% and rapid AF, AFL, or atrial tachycardia. Intravenous landiolol led to a significant reduction in HR by approximately 40% from the initial level in 29 individuals, and 9 of them experienced spontaneous termination of AF after receiving landiolol. In these patients, a higher initial LVEF was identified as a positive indicator of the effectiveness and safety of landiolol [63].

In a prospective study involving 101 patients experiencing acute decompensated heart failure classified as NYHA class IV, with an ejection fraction below 40% and associated AF, the use of landiolol was associated with a reduced risk of cardiac death, deterioration of renal function, and extended hospital stays. A better prognosis among these patients was correlated with lower left ventricular volume and high mean BP [64]. 

A recent study investigated the effectiveness of combining landiolol treatment with milrinone in nine patients experiencing acute decompensated heart failure with an ejection fraction ranging from 28% to 8%, with a “wet and cold” phenotype and rapid AF. Positive outcomes were noted with low doses of landiolol, which not only lowered HR by 11% without affecting systolic BP, but also led to a reduction in pulmonary capillary wedge pressure and an increase in stroke volume index. However, higher doses of 3 mcg/kg/min landiolol in conjunction with milrinone were linked to decreased BP, stroke volume index, and cardiac index [65]. 

### 3.4. Treatment of Sepsis-Related Atrial Fibrillation 

Cardiovascular abnormalities occurring in sepsis may be due to changes in circulating volume, venous tone, and tachycardia, all of which can lead to impaired cardiac function and changes in HR [66]. Sepsis can trigger tachyarrhythmias like AF due to heightened sympathetic system activity and elevated inflammatory cytokine levels. Once AF sets in, the prognosis for patients becomes even more uncertain [67]. An improved prognosis is associated with reducing the ventricular rate to less than 95 beats per minute within 24 h of occurrence in a patient with sepsis [66]. Several studies in the literature have highlighted the effectiveness of using the β-blocker treatment landiolol in patients experiencing AF during episodes of sepsis.

A study involving 61 septic patients admitted to the ICU with AF/AFL (39 receiving landiolol and 22 not receiving landiolol) revealed notable differences between the two study groups. With an initial dose of 6.3 ± 3.3 g/kg/min, the HR significantly decreased in the landiolol group during the first hour from 145 ± 14 bpm to 119 ± 28 bpm, with no occurrence of hypotension or severe bradycardia. Conversely, there were no changes in HR observed in the control group. In the landiolol group, 25.6% of patients converted to SR, while none in the control group returned to SR [60]. The study concludes that, in patients experiencing sepsis and AF/AFL, the administration of a low dose of landiolol can effectively decrease HR and facilitate the conversion to SR. This intervention maintains a good safety profile, avoiding the induction of hypotension or reduction in cardiac output [60]. 

In another recent multicenter, randomized study, a total of 151 patients admitted to the ICU for sepsis and subsequently developing AF or AFL during hospitalization were included. Landiolol significantly lowered the HR in 55% of patients to a range of 60–94/min within 24 h and considerably lowered the likelihood of a recurrence of arrhythmia. Additionally, among patients in whom the HR was successfully reduced, the 28-day mortality was lower (9% vs. 24%) [66]. 

The beneficial effects of β-blockers stem from their pleiotropic action, leading to the modulation of inflammation, protection of cardiomyocytes, and enhanced organ function [68]. Experimental investigations using animal models of septic shock have explored the effects of β-blockers on cardiovascular function and inflammation. The primary models utilized include polymicrobial sepsis induced by cecal ligation and puncture and endotoxic shock induced by LPS injection, which result in varying degrees of sepsis severity. Β -adrenergic blockade inhibits splenocyte apoptosis and the release of proinflammatory cytokines such as IL-6 and TNF-alpha, as well as nuclear factor -kB translocation in the early phase of septic shock [69]. Even more studies on rodents have shown that β-blockers either maintain or increase cardiac output despite a decrease in heart rate. This reduction in myocardial external work allows for higher global myocardial work efficiency at a lower energy cost [69,70,71,72]. These hemodynamic improvements correlate with a decrease in lactate levels, indicating potentially lower tissue oxygen demand and increased oxidative metabolism [69]. 

## 4. Challenges and Perspectives 

In the clinical trials currently documented in the literature, the use of landiolol was found to be safe and well-tolerated. The most frequently reported side effects after landiolol treatment are hypotension and bradycardia, which often resolved upon dose reduction or the discontinuation of treatment. This may suggest that the occurrence of adverse effects is strongly correlated with the dose of landiolol, but current studies are contradictory [67,73].

In clinical trials, the prevalence rates of hypotension in the landiolol group (*n* = 948), active comparator, no landiolol, and placebo cohorts were 8.5%, 8.5%, 5.7%, and 2.1%, respectively. The frequency of bradycardia was 2.1%, 2.5%, 2.4%, and 0%. In uncontrolled trials, hypotension and bradycardia manifested in 8.6% and 0.5% of 581 recipients of landiolol. Nevertheless, in post-marketing investigations (1257 participants), the occurrence of hypotension was documented at 0.8%, while bradycardia was observed in 0.7% of patients. All instances of hypotension and bradycardia either ameliorated or resolved spontaneously without any intervention, or within minutes after discontinuation of landiolol, with or without additional treatment [73]. 

Some studies have reported noteworthy adverse effects. For instance, in a patient experiencing substantial active bleeding, severe hypotension after landiolol treatment led to shock. The discontinuation of landiolol was necessary to improve the patient’s condition. Additional significant complications post-landiolol treatment have been documented, particularly in elderly patients or those with other cardiovascular pathologies, such as cardiac arrest and complete atrioventricular block [73]. The primary drawbacks of this β-blocker lie in its negative inotropic and chronotropic properties, which stem from reduced sympathetic activity. These properties may lead to reduced atrioventricular conduction and cardiac blockage [26].

In some cases, landiolol treatment has been responsible for worsening heart failure, or the development of pneumonia [74]. Several alterations in laboratory parameters, including elevated ALT, AST, and bilirubin, have been noted in patients treated with landiolol. However, these changes did not exhibit significant clinical manifestations and were subsequently resolved [62,73]. Other adverse effects reported in the literature are shown in Table 4.

AF can arise from multiple factors, including associated comorbidities, surgical trauma, ischemia, and reperfusion during surgery, as well as electrical and pressure changes at the atrial level. However, compelling evidence suggests that inflammation and oxidative stress play significant roles in the development of AF [78]. Landiolol showed a protective effect on LPS-induced systemic inflammation model in an animal study, being associated with a decrease in high mobility group box-1 protein (HMGB-1) and decreasing IL-6 levels, while inhibiting TNF-alpha levels [78,79]. The animal model demonstrated that landiolol’s anti-inflammatory effect extends to the inhibition of the inflammatory and vasoconstrictor peptide endothelin (ET) 1 [80]. These data emphasize the anti-inflammatory effects of landiolol, through which it helps prevent AF. However, evidence is limited, so further studies are needed to investigate this effect. 

It is important to note that the majority of studies on landiolol are conducted in Japan and are predominantly characterized by a limited number of participants. This is partly due to landiolol’s origination in Japan, where it has been extensively utilized in clinical practice. However, the focus on observational studies with smaller participant pools may pose limitations in terms of generalizability and robustness of findings. Efforts to expand research globally through larger-scale clinical trials and international collaborations are underway. These aim to provide robust evidence regarding landiolol’s efficacy, safety, and optimal use, ultimately improving patient outcomes and advancing cardiovascular care on a global scale.

## 5. Conclusions

Landiolol, a new ultra-short-acting β1 selective antagonist, shows promise in various clinical scenarios, ranging from cardiac and non-cardiac surgeries to preventing and treating AF, in the presence of heart failure or sepsis. While its ultra-short-acting nature provides advantages in specific settings, the careful consideration of potential adverse effects is crucial. Further research is needed to explore its anti-inflammatory effects and fully understand its role in preventing AF. Despite challenges, landiolol presents itself as a valuable tool in managing specific cardiac conditions in clinical practice.

## Figures and Tables

**Table 2 pharmaceutics-16-00517-t002:** An overview of Landiolol in comparison with other beta-blockers.

Property	Landiolol [26]	Esmolol [26]	Metoprolol [27]	Nebivolol [28]	Bisoprolol [28]	Atenolol [29]	Carvedilol [30]	Propranolol [31]
Drug class	Ultra-short-acting selective β-1 blocker	Short-acting selective β-1 blocker	Selective β-1 blocker	Highly cardio selective β-1 blocker with vasodilator properties	Highly cardio-selective β-1 blocker	Cardio selective β-1 blocker	Non-selective β blocker with alpha-1 blocking activity	Non-selective β blocker
Half-life	Very short (about 4 min)	Very short (about 9 min)	3–7 h	10–12 h	10–12 h	6–7 h	7–10 h	4–6 h
Pharmacokinetics	Rapid onset and offset of action	Rapid onset and offset of action	Rapidly and completely absorbed	Absorbed rapidly and extensively metabolized	Slowly and completely absorbed	Absorbed slowly but almost completely	Extensive l y metabolized	Rapidly and completely absorbed
Pharmacodynamics	Selective β1-blocker, short-acting	Selective β1-blocker, short-acting	Selective β1-blocker, long-acting	β1-blocker with vasodilator effects	Selective β1-blocker, long-acting	Selective β1-blocker, long-acting	Non-selective β-blocker with β1-blockade	Non-selective β-blocker, membrane-stabilizing activity
Cardio selectivity	Highly	Highly	Highly	Highly with vasodilator properties	Highly	Highly	Non-selective	Non-selective
Dose Range	1–40 μg/kg/min	50–300 μg/kg/min	25–200 mg/day	2.5–10 mg/day	2.5–10 mg/day	25–100 mg/day	6.25–25 mg/day	20–320 mg/day
Vasodilator effects	Minimal	Minimal	Minimal	Significant	Minimal	Minimal	Strong	Minimal
Indications	Rapid heart rate control in critical care settings	Acute heart rate control, intraoperative and postoperative tachycardia	Hypertension, angina, heart failure, post-myocardial infarction	Hypertension, angina, heart failure	Hypertension, angina, heart failure	Hypertension, angina, arrhythmias	Hypertension, angina, heart failure	Hypertension, angina, arrhythmias
Adverse effects	Rarely hypotension, bradycardia	Hypotension, bradycardia, bronchospasm	Hypotension, bradycardia, fatigue, dizziness	Hypotension, bradycardia, fatigue, dizziness, headache	Hypotension, bradycardia, fatigue, dizziness	Hypotension, bradycardia, fatigue, dizziness	Hypotension, bradycardia, fatigue, dizziness, heart block	Hypotension, bradycardia, fatigue, dizziness, bronchospasm
Contraindication	Severe bradycardia, heart block	Heart block, severe bradycardia, heart failure, asthma	Bradycardia, heart block, heart failure, hypotension	Severe bradycardia, heart block, hepatic impairment	Heart block, severe bradycardia, heart failure, asthma	Bradycar dia, heart block, heart failure, asthma	Asthma, heart block, severe bradycard ia	Asthma, heart block, bradycardia

**Table 3 pharmaceutics-16-00517-t003:** Studies that used landiolol for the treatment of post-operative atrial fibrillation in non-cardiac surgery.

Study Name	Type of Surgery	Number of Patients	Design Study	Outcomes
Nojiri et al. [49]	Lung surgery	30	Study group (*n* = 15)—landiolol 5 mcg/kg/min or 10 mcg/kg/min Control group (*n* = 15)—0.25 mg digoxin and 5 mg verapamil	Rate of conversion to SR Study group: at 2 h 8/15 (53%) at 12 h 11/15 (73%) Control group: at 2 h 3/15 (20%) at 12 h 8/15 (53%) Time of conversion to SR Study group: 8.1 ± 11.0 h Control group: 23.0 ± 26.0 h
Niwa et al. [55]	Esophagectomy	24	Study group (*n* = 11)—landiolol 6.5 ± 3.4 mcg/kg/min, increased to 7.7 ± 4.4 mcg/kg/min Control group (*n* = 13)—digoxin and calcium channel blockers.	Rate of conversion to SR Study group: at 2 h 5/8 (62.5%) at 12 h 8/8 (100%) Control group: at 2 h at 2 h 1/13 (7.7%) at 12 h 7/13 (53.8%) Time of conversion to SR Study group: 3.6 6 ± 6.6 h Control group: 23.3 ± 5.2 h
Mori et al. [54]	Esophagectomy	74	Study group (*n* = 13)—landiolol 0.01 mg/kg/min increased to 0.04 mg/kg/min	Rate of conversion to SR: 10/13 (77%)
Nakano et al. [48]	Pulmonary resection	25	Study group (*n* = 25)—landiolol 5–10 mcg/kg/min or o 1–5 mcg/kg/min	Rate of conversion to SR: 14/25 (56%)—at 1 h 4/14 (28.6%) and at 12 h 5/14 (35.7%) HR control: from 135 ± 24 bpm to 85 ± 19 bpm

SR: sinus rhythm; HR: heart rate.

**Table 4 pharmaceutics-16-00517-t004:** Adverse effects reported with landiolol treatment.

Hypotension [32,42,63,64,75,76]	Dyspnea [75]
Bradycardia [64,75,77]	Asthma [62]
Aggravation of cardiac failure [64,66,75]	Hypokalemia [74]
Ventricular tachycardia [75]	Hypoglycemia [74]
Complete atrioventricular block [64,73]	Hepatic enzyme increased [73,74]
Cardiorespiratory arrest [66,75]	Gamma-glutamyl transferase [74]
Cardiogenic shock [73,75]	Blood uric acid increased [74]
Embolic stroke [62]	Blood alkaline phosphatase increased [74]
Pneumonia [49,75]	Increased serum creatinine [64,76]
Respiratory failure [49,75]	Vomiting and nausea [62]

## Data Availability

Not applicable.

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
