# Peer review of "An Overview of the Pharmacokinetics and Pharmacodynamics of Landiolol (an Ultra-Short Acting β1 Selective Antagonist) in Atrial Fibrillation"

_pharmaceutics, 2024, doi:10.3390/pharmaceutics16040517_

Round 1

Reviewer 1 Report

Comments and Suggestions for Authors

The manuscript is well written, based on the well-selected literature, with the proper structure, style, and language; however - for the sake of bringing more useful information to the readers - it may be extended with a few sentences related to the suggestion below. 

It would be interesting to the readers to know, based on analysis of the presented studies, at which stage of operation the infusion was initiated: at the induction of anesthesia, at the cardiopulmonary bypass procedure, after the procedure, or at the ICU admission. Also, the discussion on the eventual choice of continuation therapy with other beta-blockers following the cessation of infusion would be interesting. Also the authors comment on the strategy of initiating oral beta-blocker pre-operatively and switching to the landiolol infusion intraoperatively or in the ICU merits further attention.

 This information is already available to the authors, and I believe that for additional information, the following references might be helpful [the review articles by Fellahi (2018), doi: 10.1093/eurheartj/sux038,  and Feng (2016), doi: 10.1093/eurheartj/sux038, and the original study of Yoshioka (2009), doi: 10.1055/s-0029-1186069].

 Finally, minimal wording corrections may be implemented: The text bubbled in Figure 1,“Landiolol blocking beta-1 receptor in 4 minutes” should be better worded as “Landiolol blocking beta-1 receptor for 4 minutes” for the sake of preciseness (elimination half-life).

Reviewer 2 Report

Comments and Suggestions for Authors

The authors provide a comprehensive, quite well referenced and logically presented review of the effectiveness of Landiolol a short acting beta-1 adrenergic agonist in atrial fibrillation.  As they prepare a revised version of this review, I have several overall arching and some specific requirements and recommendations.  Individually and as a collective, these are meant to clarify the text, improve the Tables and provide further referencing and documentation in support of key background material.

1.  General Comments:  It is apparent that very significant goal directed literature review and document planning has been done.  Nonetheless, there are several important areas and sections that need further referencing.  Examples include:

a)  In the text between lines 57 and approximately 94 additional References are needed concerning the selectivity and mode of action of Landiolol.  In the absence of this it seems plausible that some of the important actions of this compound would be directly on ion channels that are associated with beta receptor subsets.  

b)  The text between approximately lines 98 and 123 needs to be supported by more current and well cited reviews.  This same comment applies to your material in Table 1.  I note also, that the formatting of Table 1 needs to be improved.

c)  Regarding the text between lines 192 and 202, the key point of the selectivity of Landiolol is neither well made nor well supported.

2.  Specific Edits:  When rewriting your manuscript please replace or improve each of the following:

a) Line 27, 'approved by Japan'.  What do you mean by this?

b) Line 55, 'perfect candidate'.

c)  Line 63, 'to bring'.  I suggest 'to produce'.

d)  Line 128, I am unfamiliar with the term 'bathmotropic'.

e)  Line 135, INaK channels.  This is an unconventional and perhaps incorrect descriptor for any of the subunits of the isoforms of the sodium potassium pump integral protein that is expressed in mammalian heart.  

f)  Lines 139-141, This text offers only a dated and perhaps misleading explanation of the known mechanisms for negative inotropic effects in mammalian atria.  

g)  Lines 174-184, These two paragraphs introduce relevant and potentially interesting material.  However, neither paragraph is sufficiently well Referenced or clearly presented for it to be readable or convincing.

h)  Lines 185-191, The material in this paragraph is mainly, if not completely, redundant.  

i)  Line 305, Your term 'post-operatory' is unconventional and should be replaced.  

j)  Lines 394 - 396, Although I believe I understand this point; I suggest that its importance justifies a five sentence paragraph with additional References.

k)  Line 407, What do you mean by 'comparably lower'?

l)  Lines 418-419, The phrase 'deceleration of atrioventricular activity' is ambiguous and needs to be clarified and expanded upon.

Comments on the Quality of English Language

The authors will be able to improve the clarity and impact of this review with additional editing and some rewriting as described above.

Reviewer 3 Report

Comments and Suggestions for Authors

The present literature review focuses on Landiolol, an ultra-short acting β1 selective antagonist, that despite its favorable pharmacokinetic and pharmacodynamic properties, its utilization in large randomized controlled trials is notably lacking.

Some specific comments:

1.      In order to improve the literature review, it is important for authors to adopt a more systematic approach in referencing relevant literature. For instance, in lines 42-45 authors should provide references to support their claims about the risks of atrial fibrillation in patients with cardiovascular disease and the need for swift management leading to the development of Landiolol. Similarely in lines 152-154 references should be included to support the benefits of Landiolol in chronic obstructive pulmonary disease and in lines 196-197 regarding the approval of landiolol in Japan. This will strengthen the credibility and reliability of the literature review.

2.      Readability in Table 1 could be improved by adding some space between columns and ensuring that column “IKs” is aligned at the same level as the other three columns.

3.      It would be beneficial for authors to note that the majority of studies on Landiolol are conducted in Japan and are predominantly observational, characterized by a limited number of participants.

4.      Another consideration to be mentioned is that rate control and adverse events of landiolol are dose-dependent.

5.      It is mentioned in section 3.2 and lines 302-303 that “Landiolol demonstrated superiority over traditional therapy in effectively managing HR and facilitating the restoration of SR”, while in studies [32], [33], [38], [39], landiolol was compared only to digoxin and calcium channel blockers. None of the aforementioned antiarrhythmics are used for restoration of sinus rhythm. In two studies landiolol was not compared to other antiarrhythmic therapy. As a result, restoration of sinus rhythm should be addressed separately as a distinct point of consideration.

Comments on the Quality of English Language

1.      Additionally, there are some grammar mistakes. In line 59 authors mention “to control HR control”. The second mentioning of control should be deleted. In line 68 the word “and” can be replaced by a “,” (comma). In lines 81-82, authors mention “heart load”, but this is not a right definition. Preload, afterload or cardiac load are being most commonly used. In general there is need for extensive grammar editing

Round 2

Reviewer 2 Report

Comments and Suggestions for Authors

Thank you for considering my comments and making appropriate changes and additions.  

Comments on the Quality of English Language

Good to very good.  Journal Editors could re-read and edit if that is their policy.

Reviewer 3 Report

Comments and Suggestions for Authors

Thank you for the update. The revisions made by the authors have significantly improved the quality of the paper.  I appreciate the authors' efforts in carefully considering and incorporating the feedback provided.